# A Review of the Role of Endo/Sarcoplasmic Reticulum-Mitochondria Ca^2+^ Transport in Diseases and Skeletal Muscle Function

**DOI:** 10.3390/ijerph18083874

**Published:** 2021-04-07

**Authors:** Shuang-Shuang Zhang, Shi Zhou, Zachary J. Crowley-McHattan, Rui-Yuan Wang, Jun-Ping Li

**Affiliations:** 1School of Sport Science, Beijing Sport University, Beijing 100084, China; s.zhang.27@student.scu.edu.au (S.-S.Z.); doctorljp@126.com (J.-P.L.); 2Faculty of Health, Southern Cross University, East Lismore, NSW 2480, Australia; shi.zhou@scu.edu.au (S.Z.); zac.crowley@scu.edu.au (Z.J.C.-M.)

**Keywords:** mitochondria-associated membrane, endo/sarcoplasmic reticulum-mitochondria Ca^2+^ transport, mitochondrial calcium overload, skeletal muscle function

## Abstract

The physical contact site between a mitochondrion and endoplasmic reticulum (ER), named the mitochondria-associated membrane (MAM), has emerged as a fundamental platform for regulating the functions of the two organelles and several cellular processes. This includes Ca^2+^ transport from the ER to mitochondria, mitochondrial dynamics, autophagy, apoptosis signalling, ER stress signalling, redox reaction, and membrane structure maintenance. Consequently, the MAM is suggested to be involved in, and as a possible therapeutic target for, some common diseases and impairment in skeletal muscle function, such as insulin resistance and diabetes, obesity, neurodegenerative diseases, Duchenne muscular dystrophy, age-related muscle atrophy, and exercise-induced muscle damage. In the past decade, evidence suggests that alterations in Ca^2+^ transport from the ER to mitochondria, mediated by the macromolecular complex formed by IP_3_R, Grp75, and VDAC1, may be a universal mechanism for how ER-mitochondria cross-talk is involved in different physiological/pathological conditions mentioned above. A better understanding of the ER (or sarcoplasmic reticulum in muscle)-mitochondria Ca^2+^ transport system may provide a new perspective for exploring the mechanism of how the MAM is involved in the pathology of diseases and skeletal muscle dysfunction. This review provides a summary of recent research findings in this area.

## 1. Introduction

Mitochondria and endoplasmic reticulum (ER) are two organelles in a cell that play essential roles in cellular survival and stress responses. The mitochondrion is the powerhouse of the cell. The ER is involved in protein synthesis, modification, secretion, lipid and steroid synthesis, and modulation of Ca^2+^ signalling. Sarcoplasmic reticulum (SR) is a special form of ER in striated muscle; it is specialised in the lipid synthesis, Ca^2+^ modulation, and excitation–contraction coupling. Structurally and functionally, these two organelles are connected via the mitochondria-associated membrane (MAM), a contact site that acts as a platform for inter-organelle communication [1]. In particular, the MAM is involved in a multitude of cellular processes [2], among which the Ca^2+^ transport from the ER to the mitochondrion (ER-mitochondria Ca^2+^ transport) is the focus of this review. This process is mediated by the molecular complex inositol 1,4,5-triphosphate receptor (IP_3_R)–glucose-regulated protein 75 (Grp75)–voltage-dependent anion channel 1 (VDAC1), which works as a hub for cellular survival and death based on its role in regulating mitochondrial physiology and cellular Ca^2+^ homeostasis [3,4]. During the past decade, alterations of ER-mitochondria Ca^2+^ transport have been observed in diabetes, obesity, cancer, neurodegenerative diseases like Alzheimer’s disease and Parkinson’s disease, and ischemia/reperfusion injuries in which the integrity or number of MAMs are changed [2,5].

Skeletal muscle is a high energy-consuming organ in which adequate energy supply is essential for muscle performance. Changes in mitochondrial Ca^2+^ homeostasis could cause Ca^2+^-dependent damage to mitochondrial structure and function. Conjecture exists that these changes contribute to the age- and disease-related attenuation in muscle performance [6]. There are also some indications that these changes may contribute to exercise-induced muscle damage [7,8,9]. The [Ca^2+^] variations on either the outer mitochondrial membrane (OMM) surface or within mitochondria ([Ca^2+^]_mt_), in response to stimulation of the IP_3_R, are frequently measured for exploring the Ca^2+^ communication between the ER and mitochondria in the liver and brain [10,11,12]. However, few studies have been undertaken on skeletal muscle in this regard. The MAM’s role concerning impaired skeletal muscle function has been confirmed in muscular dystrophy, ageing, and insulin resistance, with the evidence that there is a reduction in the number of MAMs in these conditions [13,14,15]. A reduced MAM number has also been implicated in heavy-load endurance exercise that may be caused by SR stress and an increased mitochondrial division rate [16,17,18]. Furthermore, according to the studies investigating MAM and ER-mitochondria Ca^2+^ transport in response to ER stress [13], mitochondrial dynamics [19], and insulin resistance [20], speculation has arisen that this transport system plays an essential role in mitochondrial Ca^2+^ overload in response to intensive exercise (Figure 1).

The aims of this review are (1) to summarise the current literature on the function of the MAM and Ca^2+^ transport from the ER (or SR) to mitochondria, mediated by the IP_3_R-Grp75-VDAC1 complex, in regulating mitochondrial Ca^2+^ homeostasis; and (2) to discuss the changes in the MAM and ER/SR-mitochondria Ca^2+^ transport concerning diseases and skeletal muscle function.

## 2. Mitochondria-Associated Membrane: A Platform for ER-Mitochondria Ca^2+^ Transport 

The calcium ion (Ca^2+^) in the mitochondrial matrix is required to stimulate oxidative metabolism by regulating three rate-limiting enzymes in the tricarboxylic acid cycle (TCA): isocitrate dehydrogenase, α-ketoglutarate dehydrogenase, and pyruvate dehydrogenase. It works as a booster in the synthesis of the reduced oxidative phosphorylation substrates (NADH and FADH_2_) [21]. Physiologically, mitochondrial Ca^2+^ accumulation is required for enhancing mitochondrial bioenergetics and cellular processes in response to cell stress. However, an excessively high [Ca^2+^]_mt_, i.e., mitochondrial Ca^2+^ overload, could cause a constant opening of the mitochondrial permeability transition pore (mPTP). This may subsequently lead to mitochondrial swelling, rupture of the OMM, and the collapse of the mitochondrial membrane potential, resulting in structural and functional damage to mitochondria [22].

### 2.1. High Ca^2+^ Microdomain between the ER and Mitochondria 

Under resting conditions, the OMM [23] is permeable to ions and small proteins. Predominantly this is due to the facilitation by the voltage-dependent anion channel 1 (VDAC1) for the diffusion of succinate, pyruvate, ATP, etc., across the OMM [24]. The voltage-dependent anion channel is also permeable to some ions, e.g., K^+^, Na^+^, Ca^2+^, and it has higher permeability to Ca^2+^ in a high potential state (so-called “closed” state) than in a low potential state (so-called “opened” state). It has been reported that the VDAC1 possesses Ca^2+^-binding sites, and its closure promotes Ca^2+^ influx into mitochondria, which causes the opening of the mitochondrial permeability transition pore. Consequently, closure of the VDAC1 is recognised as a pro-apoptotic signal [25]. At the physiological state, [Ca^2+^] in the mitochondrial intermembrane space is indistinguishable from that in the cytoplasm ([Ca^2+^]_c_) (around 100 nM) [26]. Mitochondrial Ca^2+^ uptake is driven by the electrochemical gradient across the inner mitochondrial membrane (IMM) (−150 to −180 mV, negative inside) [27], while the mitochondrial calcium uniporter (MCU) on the IMM is responsible for mitochondrial Ca^2+^ uptake. The MCU is characterised by a low Ca^2+^ affinity (KD = 20–30 μM), which makes it hard to understand how mitochondria uptake Ca^2+^ at such a low level of [Ca^2+^] under resting conditions or even in response to physiological stimulation (approximately 1 μM) [28]. This paradox is explained by the widely accepted “Ca^2+^ microdomain” hypothesis, in which mitochondrial Ca^2+^ accumulation is effectively favoured by the high Ca^2+^ microdomain close to the ER Ca^2+^ release channels [29]. The [Ca^2+^] at the ER–mitochondrial interface is much higher than that in bulk cytoplasm. This unique [Ca^2+^] localisation has been directly demonstrated by monitoring the local [Ca^2+^] at the ER–mitochondrial interface by fluorescent interorganelle linkers [10], or GFP-based Ca^2+^ probe localised on the cytosolic surface of the OMM [29]. Upon the Ca^2+^ mobilisation from the ER, the [Ca^2+^] in local regions near the mitochondrial surface is 5–10-fold higher than that in bulk cytoplasm [30]. The local [Ca^2+^] at the ER–mitochondrial interface may then rise to several tens of micromolar during the activation of the IP_3_R. There are two important types of Ca^2+^ release channels at the ER membrane, ryanodine receptor (RyR) and IP_3_R. While IP_3_R is mainly for forming the high Ca^2+^ microdomain [11,12,31], the RyR has a major role in the excitation–contraction coupling processe in skeletal muscle [32]. 

### 2.2. Endoplasmic Reticulum-Mitochondrial Ca^2+^ Transport 

The IP_3_R is predominantly localised on the ER membrane and is one of the important Ca^2+^ release channels of ER. There are three IP_3_R isoforms, IP_3_R1, IP_3_R2, and IP_3_R3 in mammalian cells. The ion permeability of the IP_3_R could be regulated by inositol 1,4,5-triphosphate (IP_3_), cytosolic Ca^2+^, and ATP. Upon receiving a stimulus, IP_3_ is produced by the cleavage of phosphatidylinositol 4,5-bisphosphate (PIP2). In response to the presence of IP_3_, the IP_3_R releases Ca^2+^ from the ER [33]. The ER is a major organelle for storage of intracellular Ca^2+^. The efficiency of mitochondrial Ca^2+^ uptake depends on the close localisation of mitochondria to Ca^2+^ release channels at the ER membrane that is essential to Ca^2+^ diffusion from the ER to mitochondria (ER-mitochondria Ca^2+^ transport) [29]. The IP_3_R is required for increasing [Ca^2+^]_mt_, and at times activation of this protein can result in [Ca^2+^]_mt_ 20 times higher than the global increase in [Ca^2+^]_c_ [31]. Additionally, the activation of IP_3_R may evoke a large [Ca^2+^] change at the ER–mitochondrial interface even when the increase in [Ca^2+^]_c_ is suppressed in bulk by the ethylene glycol-bis(β-aminoethyl ether)-N,N,N’,N’-tetraacetic acid (EGTA)/Ca^2+^ buffer [10]. These findings suggest that IP_3_R plays a vital role in mediating ER-mitochondria Ca^2+^ transport. The VDAC1 protein is also required for the control of ER-mitochondria Ca^2+^ transport. Recombinant expression of the VDAC1 could enhance the transfer of Ca^2+^ from the microdomains to mitochondria [34]. Moreover, a recombinant expression of IP_3_R-LBD_224–605_ increases the ER-mitochondria Ca^2+^ transport, and this phenomenon can be inhibited by the downregulation of Grp75 [35]. The Grp75 is a 75 kDa glucose-regulated protein and molecular chaperone that belongs to the heat shock protein 70 (Hsp70) family. Predominantly, Grp75 is localised in the mitochondrial matrix, while it is also found in cytoplasm or other subcellular compartments [35]. This protein is demonstrated to tether the N-terminal domain of the IP_3_R to VDAC1 and enhance mitochondrial Ca^2+^ accumulation by stabilising the conformation or coupling of IP_3_R and VDAC1 [36].

The physical link between the ER and mitochondria is necessary for normal ER-mitochondrial Ca^2+^ transport, with the [Ca^2+^]_mt_ being increased by moving the ER closer to the mitochondria [37]. Additionally, the ER-mitochondria Ca^2+^ transport may depend on an appropriate distance between the ER and mitochondria. One study reported that a distance of ~20 nm or shorter between the two organelles is required for proper ER-mitochondria Ca^2+^ transport [30]. Although the optimal or average width of the MAM that coordinates efficient ER-mitochondria Ca^2+^ transport is still unknown, it is hypothesised that the width can impact the Ca^2+^ transfer by regulating the assembly of the IP_3_R-Grp75-VDAC1 complex [30]. 

During the past decade, the function of the IP_3_R-Grp75-VDAC1 complex in transporting Ca^2+^ has attracted a great deal of attention [4,13,38]. Numerous proteins, like DJ-1 [39], transglutaminase type 2 (TG2) [35], FUN14 domain-containing protein 1 (FUNDC1) [19], glycogen synthase kinase-3β (GSK3β) [5], pyruvate dehydrogenase kinase 4(PDK4) [15], etc., are found to regulate the integrity of MAM, or interact with the complex IP_3_R-Grp75-VDAC1, influencing Ca^2+^ transport from ER to mitochondria. Abnormal transmission of Ca^2+^ between the ER and mitochondria has been implicated in several pathological conditions [2,5,40,41,42,43]. According to the current literature, maintaining the structural integrity of the MAM and ER-mitochondria Ca^2+^ transport within normal physiological ranges may be essential to retarding either the pathogenesis or the progression of these diseases.

## 3. Effects of the MAM and SR-Mitochondria Ca^2+^ Transport on Skeletal Muscle Function

Over the past 40 years, the idea that mitochondrial dysfunction contributes to muscle function impairment has been hypothesised and tested [6]. It is now understood that the disruption of mitochondrial Ca^2+^ homeostasis is a general mechanism for mitochondrial function impairment [44]. Mitochondria-associated membranes were found in skeletal muscle about 30 years ago [45], adding to the evidence that the MAM contributes to regulating mitochondrial Ca^2+^ homeostasis. As the SR is a large Ca^2+^ store regulating intracellular and mitochondrial Ca^2+^ concentrations, the Ca^2+^ cross-talk between SR and mitochondria has attracted increased attention in the recent past [23]. 

### 3.1. Mitochondria-Associated Membrane in Cardiac and Skeletal Muscle 

Close localisation between the SR and mitochondria was first demonstrated in frog skeletal muscle using scanning electron microscopy in 1987 [45]. More recently, Ainbinder et al. (2015) found that mitochondria were located close to the I-band in the flexor digitorum of mice, and that the OMM was linked to the terminal cistern of SR by MAMs. They additionally found that the MAMs in the skeletal muscle were established and maintained during postnatal maturation. At the same time, mitochondria shifted from a longitudinal cluster to a transversal orientation and moved from the H-band to the I-band [46]. Furthermore, protein analysis showed that SR membrane fractions contained proteins involved in oxidative phosphorylation and the TCA [47]. These findings indicate that these proteins in mitochondria are connected with proteins in the cell’s SR, and that the SR and mitochondria are also functionally connected through MAMs [47]. 

There is currently limited experimental research demonstrating the molecular composition of SR–mitochondria interaction and its specific functions in skeletal muscle. However, numerous studies have investigated MAM formation proteins and their functional relevance in the brain, liver, and cardiac muscle [48,49,50]. A protein analysis of the MAM in skeletal muscle has identified 459 proteins, including 101 membrane proteins, in which VDAC1, VDAC2, and VDAC3 might participate in the formation of the tethers between SR and mitochondria [51]. Among the MAM constitutive proteins, mitofusin 2 (Mfn2) is most commonly recognised as a molecule for regulating the structure of the MAMs [52,53]. However, in other studies, Mfn2 is demonstrated to be a physical tether protein [54] or a tethering antagonist [55]. In skeletal muscle, Mfn2 may work as a tether protein because its depletion leads to mitochondrial migration away from the I-band towards the A-band, resulting in reduced MAMs [46]. Moreover, proteins that are involved in Ca^2+^ transport from the ER to mitochondria, like IP_3_R and Grp75, have been attracting increased investigative attention in the recent past. Additionally, the investigations suggest these ‘emerging factors’ being directly involved in the changes of mitochondrial Ca^2+^ homeostasis [15,56].

### 3.2. Ca^2+^ Release by SR in Skeletal Muscle Fibres

It is well known that Ca^2+^ is essential in the excitation–contraction coupling process during muscle contraction [57]. Calcium ions required for skeletal muscle contraction are predominantly released from the SR via the Ca^2+^ release unit (CRU). The CRU consists of the L-type Ca^2+^ channels (dihydropyridine receptors, DHPRs) on the exterior membrane and the RyRs on the SR [58]. The major type of RyR in skeletal muscle is RyR1. The RyR1 on the SR membrane is positioned facing the voltage-dependent L-type Ca^2+^ channels (Cav1.1) that are located on the transverse tubules (T-tubules, Figure 2). This position is the key factor for Ca^2+^ being released by the RyR upon T-tubule membrane depolarisation, causing a conformational change of the DHPRs [59]. 

Furthermore, Ca^2+^ is also required by the mitochondrial matrix in the energy (ATP) supplying processes for muscle contraction [60]. These processes require Ca^2+^ to be under precise physiological control within the mitochondria of skeletal muscle [61]. In skeletal muscle, mitochondria are located close to the CRU [14]. This localisation might be due to the inhibitory effect of Ca^2+^ on mitochondria motility so that mitochondria are kept around the Ca^2+^ intracellular stores [62]. In addition to Ca^2+^ release by RyR’s from the SR to the cytosol, the cellular membrane depolarisation could also cause Ca^2+^ release via the IP_3_R protein [63]. The Ca^2+^ cross-talk between SR and mitochondria was demonstrated by comparing the [Ca^2+^]_mt_ with the [Ca^2+^]_c_ in response to caffeine when using a fast Ca^2+^ buffer BAPTA (1, 2-bis(*2*-aminophenoxy)ethane-N, N, Nʹ, Nʹ-tetraacetic acid) in the cytosol, with the mitochondrial Ca^2+^ concentration increasing and cytosolic Ca^2+^ showing little change [64]. This finding indicates that the Ca^2+^ could be transported from the SR to mitochondria directly, not depending on the increased cytosolic Ca^2+^ concentration. According to Csordás et al. (2010), Ca^2+^ cross-talk between the SR and mitochondria is likely caused by activation of IP_3_R and depends on the close location between the SR and mitochondria. It was found that mitochondrial Ca^2+^ concentration changes in skeletal muscle may depend on the mitochondrial location being close to the CRU’s [65]. Furthermore, overexpression of VDAC1 can increase Ca^2+^ transport into mitochondria from the high Ca^2+^ microdomain around SR [34]. 

It is known that Ca^2+^ can be transported from the ER/SR to mitochondria via the IP_3_R-Grp75-VDAC1 complex in cardiac muscle and other types of cells. However, whether this is a major pathway and how the mitochondrial and cellular function is affected by this pathway in skeletal muscle are currently not clear and require further investigation. 

## 4. Sarcoplasmic Reticulum-Mitochondrial Ca^2+^ Transport in Relation to Skeletal Muscle Dysfunction

Mitochondria-associated membranes have been shown to affect the metabolic adaption and cellular function in response to physical environmental changes [66]. Evidence suggests that changes in the MAM and ER/SR-mitochondria Ca^2+^ transport are related to several health conditions and impaired skeletal muscle function, such as insulin resistance [40], Duchenne muscular dystrophy (DMD), or age-related muscle dystrophy [67]. Insulin resistance is a metabolic disorder, and the MAM has been shown to be a hub for insulin resistance in three key insulin-targeting tissues or cells: the liver [68], adipocytes [69], and skeletal muscle [70]. Reorganisation of the MAMs and mitochondrial Ca^2+^ overload has been shown in the liver of obese animals. A downregulation of MAMs and the ER-mitochondria Ca^2+^ transport has been demonstrated to improve glucose metabolism [71].

Additionally, it has been found that a reduction in ER-mitochondria Ca^2+^ transport may contribute to insulin resistance in liver cells and adipocytes [40,72]. The disruption of the MAMs in hepatocytes plays a central role in developing insulin resistance [72]. This was demonstrated by using sulforaphane for treating insulin resistance, with insulin resistance being improved along with a reduced MAM disruption and impaired interaction between VDAC1 and IP_3_R1 [56]. Moreover, in adipose tissue of obese mice and humans who were suffering from type 2 diabetes mellitus, [Ca^2+^]_mt_ was found to be increased, but [Ca^2+^]_c_ showed no change, with an accompanying rise in MCU expression and activity [73]. 

Skeletal muscle is an important regulator for whole-body glucose homeostasis and has been extensively studied concerning insulin resistance pathologies [74]. Tubbs et al. (2018 b) showed that the gastrocnemius muscle of obese mice with type 2 diabetes exhibited a significant disruption of SR–mitochondria interaction, which is compatible with experimental findings in human patients with obesity and type 2 diabetes [70]. These patients showed a reduction in VDAC1/IP_3_R interaction, indicating that the Ca^2+^ communication between the SR and mitochondria is likely to play a key role in altered glucose homeostasis in type 2 diabetes [70]. One study by Zhe et al. (2020) investigated the improvement of glucose intolerance induced by swimming. The findings suggested that decreased MAM quantity might help in understanding the mechanism of type 2 diabetes mellitus [75]. However, contrary to the findings mentioned above, there has been evidence suggesting that the increased formation of MAMs is a contributor to insulin resistance. For example, one study found that mitochondrial Ca^2+^ accumulation in skeletal muscle with insulin resistance may be caused by augmented MAM formation [15]. This is because insulin signalling can be improved by preventing MAM-mediated mitochondrial Ca^2+^ accumulation and mitochondrial dysfunction through inhibiting pyruvate dehydrogenase kinase 4 [15]. The opening of mitochondrial permeability transition pores is a necessary contributor to skeletal muscle insulin resistance because it prevents the insulin-induced translocation of glucose transporter 4 into the cytoplasm [76]. Additionally, mitochondrial Ca^2+^ overload is commonly regarded as a central contributor to the opening of mitochondrial permeability transition pores [6,24]. As shown above, controversy still exists regarding the relationship between insulin resistance and changes in the MAMs, SR-mitochondria Ca^2+^ transport, and mitochondrial Ca^2+^ homeostasis in skeletal muscle. It is also far from clear whether the alterations in MAM and IP_3_R-Grp75-VDAC1 result from or are the cause of muscular insulin resistance. 

Muscle weakness and wasting have long been confirmed as pathogenic DMD symptoms [77]. It is accepted that the defective gene dystrophin is involved in the DMD pathology [78], and it has been suggested that the increase in Ca^2+^ release by the sarcoplasmic reticulum is involved in calcium dysregulation in DMD myotubes. In DMD foetal muscle, myopathology begins with a significant delay of foetal muscle differentiation at an early foetal stage. This phenomenon is related to the activation of the PLC/IP_3_R/calcium/PKCα pathway, which may be induced by the absence of dystrophy [79]. An increased IP_3_R activity negatively regulates autophagy and elevates the [Ca^2+^]_c_ and [Ca^2+^]_mt_ [80]. However, the increase in IP_3_R expression does not enhance mitochondrial Ca^2+^ uptake because of the reduction of the complex quantity of IP_3_R-Grp75-VDAC1 [80]. Thus, the reduced ER-mitochondria Ca^2+^ transport is a suspected contributor to the reduced [Ca^2+^]_mt_ in DMD [13]. 

Disrupted SR-mitochondria Ca^2+^ transport may also potentially play a role in muscle dysfunction in ageing through mitochondrial Ca^2+^ homeostasis impairment. Further, tetanic stimulation on the flexor digitorum brevis of mice was reportedly the cause of reduced mitochondrial Ca^2+^ uptake, along with a decrease in the association between mitochondria and CRUs in aged mice [14]. In aged mice with knockdown of mitofusin 2, repetitive high-frequency tetanic stimulation reduced an impaired association between CRUs and mitochondria. This was caused by a shift of mitochondria away from the triadic positions, which may explain the declined muscle performance associated with ageing [14]. Gill et al. (2019) reported that the SR-mitochondria Ca^2+^ transport proteins IP_3_R1, Grp75, and VDAC were downregulated and that the IP_3_ gene expression reduced in the skeletal muscle of aged mice [81]. An ameliorated age-dependent skeletal muscle dysfunction was reported to be related to a reduced SR Ca^2+^ release and increased SR Ca^2+^ load [82]. Exercise-induced muscle function improvement is hypothesised to be attributed to the increase in [Ca^2+^]_mt_ caused by the increased MCU expression [83], which is probably related to the improved/maintained association between CRUs and mitochondria [84]. Based on these findings, the increasing loss of Ca^2+^ homeostasis in the SR and mitochondria are said to cause the functional decline of skeletal muscle in ageing. The defective SR-mitochondria Ca^2+^ cross-talk may precede muscle function impairment in age-related dystrophy, but further studies on molecular mechanism and Ca^2+^ signalling are required to understand this explanation better.

## 5. Sarcoplasmic Reticulum-Mitochondria Ca^2+^ Transport and Mitochondrial Ca^2+^ Overload

It is commonly accepted that ER stress, which can be caused by ageing, genetic mutations, or environmental factors, can result in various diseases, including those mentioned above [85]. The increased ER Ca^2+^ release through Ca^2+^ channels is an essential cause of ER stress, and ER stress can be attenuated by the cone-specific gene deletion of IP_3_R1 [86]. The disturbance of mitochondrial Ca^2+^ homeostasis, mostly through mitochondrial Ca^2+^ overload, also occurs in the diseases mentioned above, and it is probably due to the increased ER-mitochondria Ca^2+^ transport. In the early phase of ER stress, mitochondria were found to migrate to the ER’s periphery [18,87], while in the later phase, mitochondrial Ca^2+^ overload was shown to be caused by mitochondrial migration to the ER surface, resulting in mitochondria-dependent apoptosis [88]. In addition, according to a recent study on macromolecular complex IP_3_R-Grp75-VDAC1, Ca^2+^ transport from the ER to the mitochondria is likely to contribute to podocyte apoptosis by facilitating mitochondrial Ca^2+^ overload [4]. As can be seen from the evidence discussed above, the Ca^2+^ transport from the ER to the mitochondria may contribute to the ER stress and mitochondrial Ca^2+^ overload at the same time, but this speculation needs further experimental evidence. 

## 6. Conclusions 

This review focused on the mitochondria-associated membrane’s role, particularly the IP_3_R-Grp75-VDAC1 pathway, in calcium ion transport from the endoplasmic/sarcoplasmic reticulum to mitochondria, in relation to normal and abnormal cellular functions. Current evidence suggests that the distance between the ER/SR and mitochondria, and the modulation of the IP_3_R, Grp75, and VDAC1 activities, play a critical role in regulating mitochondrial Ca^2+^ homeostasis. A disruption in this pathway and subsequently Ca^2+^ homeostasis (such as mitochondrial Ca^2+^ overload or insufficient mitochondrial Ca^2+^) is believed to be a common causative factor in some health conditions, such as neurodegenerative diseases, obesity, insulin resistance, cancer, and ischemia/reperfusion damage, as well as in age-related muscle dystrophy and intensive exercise-induced reduction of skeletal muscle function. However, the exact cause-and-effect relationship and the potential physiological/pathological mechanisms under specific conditions require further investigation to shed light on potential therapeutic targets or interventions for such conditions. 

## Figures and Tables

**Figure 1 ijerph-18-03874-f001:**
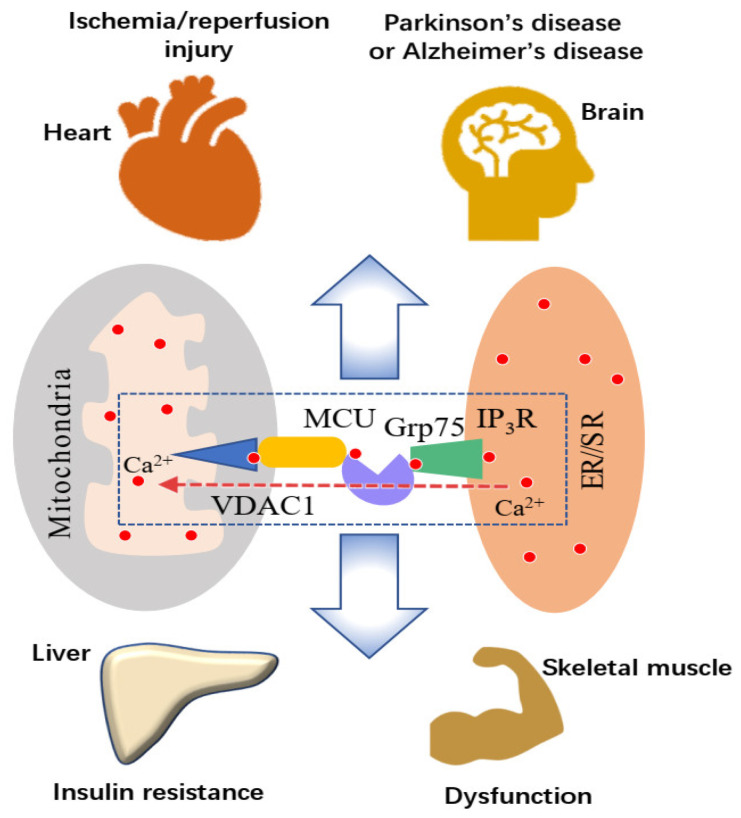
The changes in ER/SR-mitochondrial Ca^2+^ transport mediated by IP_3_R-Grp75-VDAC1 that are involved in diseases and skeletal muscle dysfunction. Under normal physiological conditions, parts of the mitochondrial membrane are physically connected to the ER/SR. The ER/SR-mitochondria Ca^2+^ transport is mediated by the macromolecular complex IP_3_R-Grp75-VDAC1 and supplies Ca^2+^ to mitochondria for ATP production. However, an increase in ER/SR-mitochondria Ca^2+^ transport is involved in diseases such as Alzheimer’s disease, hepatic diabetes, heart ischemia/reperfusion injury, and skeletal muscle impairment. A decrease in the ER/SR-mitochondria Ca^2+^ transport is also a contributing factor for Parkinson’s disease, insulin resistance in the liver, and muscle performance reduction.

**Figure 2 ijerph-18-03874-f002:**
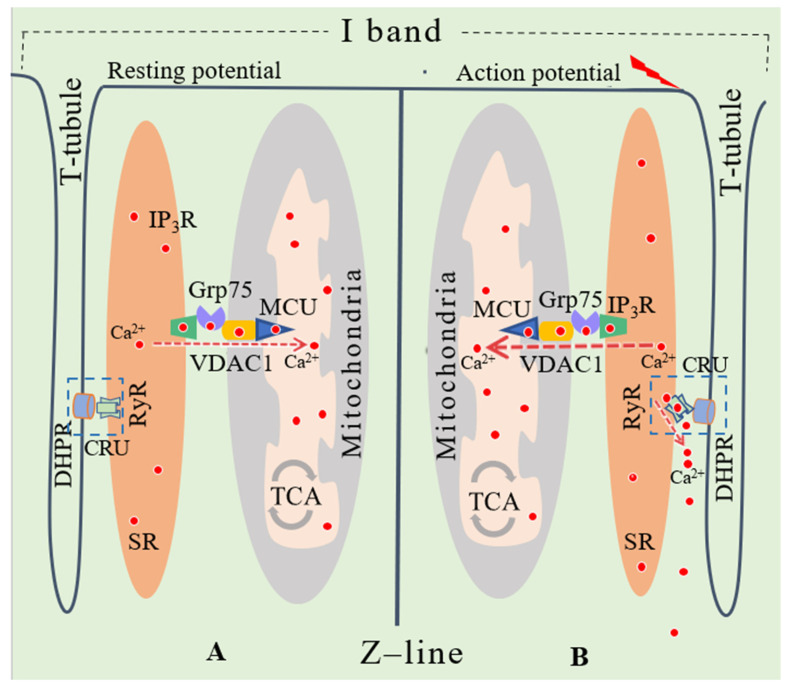
SR-mitochondria Ca^2+^ transport in skeletal muscle. The terminals of the SR are located near the T-tubules. Mitochondria are located close to the CRU on both sides of the Z-line within the I-band. (**A**) At rest, RyR releases no Ca^2+^. (**B**) Upon membrane depolarization, the conformational change of RyR causes Ca^2+^ release from the SR to cytosol (triggering muscle contraction) and hypothetically transported to mitochondria via the IP_3_R-Grp75-VDAC1 pathway (required by ATP production).

## Data Availability

Not applicable.

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
