# Peer review of "A Review of the Role of Endo/Sarcoplasmic Reticulum-Mitochondria Ca2+ Transport in Diseases and Skeletal Muscle Function"

_ijerph, 2021, doi:10.3390/ijerph18083874_

Round 1

Reviewer 1 Report

A review article by Zhang et al. concisely summarize the role of calcium transport cross-talk between ER/SR and mitochondria. Particularly, authors focus on IP3 receptor-Grp75- VDAC1 signaling in this mechanism. The manuscript is well written in the cell biology aspect. If it is elaborated in areas of molecular structure of the signaling proteins, the paper will be more comprehensive and informative. Below are some examples.

A section describing molecular, structure, and functional details of three proteins (IP3R, Grp75, VDAC) would be helpful.

Have any human disease-linked mutations in these proteins reported?

Recently, a new member, DJ-1, was proposed in this signaling complex in regard of Parkinson’s disease. Including this may make the manuscript a timelier review article.

Author Response

Point 1: A section describing molecular, structure, and functional details of three proteins (IP3R, Grp75, VDAC) would be helpful.

Response 1: Thank you for the suggestion. In accordance with the aims of our review, we have added related contents about the molecular and functional details of VDAC1 (in lines 86-91), IP3R (in lines 111-116) and Grp75 (in lines 127-131) in different paragraph.

Point 2: Have any human disease-linked mutations in these proteins reported?

Response 2: Currently, there is no article about disease-linked mutations in these proteins.

Point 3: Recently, a new member, DJ-1, was proposed in this signaling complex in regard of Parkinson’s disease. Including this may make the manuscript a timelier review article.

Response 3: Thanks for your suggestion. During the past 10 years, numerous proteins, like DJ-1, transglutaminase type 2 (TG2), FUN14 domain containing 1 (FUNDC1), Glycogen synthase kinase-3β (GSK3β), pyruvate dehydrogenase kinase 4(PDK4), etc. are found to regulate the integrity of MAM, or interact with the complex IP3R-Grp75-VDAC1, influencing Ca2+ transport from ER to mitochondria. Some of these proteins were reported to be involved in diseases, reperfusion injury, insulin resistance, Parkinson’s disease, etc. For our review, we mainly want to emphasise the certain role of ER-mitochondrial Ca2+ transport mediated by the IP3R-Grp75-VDAC1 complex, in diseases and skeletal muscle dysfunction. The information on the related molecules or proteins involved in regulating the ER-mitochondrial Ca2+ transport complex would require a significant addition to the text. We would keep the review more focused, but thank you again for the suggestion.

Reviewer 2 Report

This review describes the role of the ER/SR-mitochondrion Ca2+ transport in diseases and skeletal muscle function. This MS could represent a very important review for people working in this field. It is clear, well documented and illustrated.

I have got only few concerns and remarks:

. VDAC1: this transporter is very important. However, its name means that it transports anion, but a reader coming from outside the mitochondrion field could be surprised that it can also transport Ca2+ ions. Maybe the authors should write a specific paragraph about VDAC1 ?

. Lines 17 and 35: written like that it is difficult to understand what is this complex. The authors should precise that this complex “is formed by IP3R, Grp75 and VDAC1”

. Line 30: ER is well known to play an important role in protein synthesis, is it the same for SR ?

. Line 198: “the cellular membrane depolarization could also cause Ca2+ release via IP3R protein”. IP3R are not voltage-gated channels, so no release can be due to IP3R activation by a membrane depolarization…

. In the 2 figure legends, “this is a figure” is written…

Author Response

Point 1: VDAC1: this transporter is very important. However, its name means that it transports anion, but a reader coming from outside the mitochondrion field could be surprised that it can also transport Ca2+ ions. Maybe the authors should write a specific paragraph about VDAC1 ?

Response 1: Thanks. This problem was also raised by another reviewer. We added the necessary contents (in lines 86-91) for clarifying VDAC1 function in transporting Ca2+.

Point 2: Lines 17 and 35: written like that it is difficult to understand what is this complex. The authors should precise that this complex “is formed by IP3R, Grp75 and VDAC1”

Response 2: Thanks for your suggestion. For an easier understanding the complex formation, we have made the description for the complex more precise in lines 16-19. For the first description of the complex in the main text, we think it is better to use the full name of each of the proteins, rather than using their abbreviations.

Point 3: Line 30: ER is well known to play an important role in protein synthesis, is it the same for SR ?

Response 3: It is commonly known that ER is involved in protein synthesis, modification, secretion, lipid and steroid synthesis and modulation of Ca2+ signaling. Sarcoplasmic reticulum (SR) is a special form of ER in striated muscle, it is specialised in the lipid synthesis, Ca2+ modulation and excitation-contraction coupling. These characteristics are introduced in lines 29-32 in our revised review.

Point 4: Line 198: “the cellular membrane depolarization could also cause Ca2+ release via IP3R protein”. IP3R are not voltage-gated channels, so no release can be due to IP3R activation by a membrane depolarization…

Response 4: It is commonly known that IP3R is the Ca2+ channel that can be activated by its combination with IP3. In 2018, an article [1] reported that both the RyR and IP3R activation could be induced by membrane depolarization, either of their activation can contribute to mitochondrial Ca2+ accumulation.

Point 5:  In the 2 figure legends, “this is a figure” is written…

Response 5: We formatted our review according to the “ijerph-template” word document downloaded from the IJERPH offical website. In this document, there is an illustration as “this is a figure” under the inserted figures, which made us confused. We deleted “this is a figure” in our revised manuscript.

  1. Díaz-Vegas, A.R., et al., Mitochondrial Calcium Increase Induced by RyR1 and IP3R Channel Activation After Membrane Depolarization Regulates Skeletal Muscle Metabolism. Frontiers in Physiology, 2018. 9: p. 791-805.